

# Induced pluripotent stem cells from human hair follicle keratinocytes as a potential source for *in vitro* hair follicle cloning

Sheng Jye Lim[1], Shu Cheow Ho[1], Pooi Ling Mok[2,3], Kian Lee Tan[1], Alan H.K. Ong[1] and Seng Chiew Gan[1]

[1] Faculty of Medicine and Health Sciences, Universiti Tunku Abdul Rahman, Bandar Sungai Long, Selangor, Malaysia
[2] Department of Biomedical Sciences, Faculty of Medicine and Health Sciences, Universiti Putra Malaysia, Serdang, Selangor, Malaysia
[3] Genetics and Regenerative Medicine Research Centre, Faculty of Medicine and Health Sciences, Universiti Putra Malaysia, Serdang, Selangor, Malaysia

## ABSTRACT

**Background**. Human hair follicles are important for the renewal of new hairs and their development. The generation of induced pluripotent stem cells (iPSCs) from hair follicles is easy due to its accessibility and availability. The pluripotent cells derived from hair follicles not only have a higher tendency to re-differentiate into hair follicles, but are also more suited for growth in hair scalp tissue microenvironment.

**Methods**. In this study, human hair follicular keratinocytes were used to generate iPSCs, which were then further differentiated *in vitro* into keratinocytes. The derived iPSCs were characterised by using immunofluorescence staining, flow cytometry, and reverse-transcription PCR to check for its pluripotency markers expression.

**Results**. The iPSC clones expressed pluripotency markers such as TRA-1-60, TRA-1-81, SSEA4, OCT4, SOX2, NANOG, LEFTY, and GABRB. The well-formed three germ layers were observed during differentiation using iPSCs derived from hair follicles. The successful formation of keratioctyes from iPSCs was confirmed by the expression of cytokeratin 14 marker.

**Discussion**. Hair follicles represent a valuable keratinocytes source for *in vitro* hair cloning for use in treating hair balding or grafting in burn patients. Our significant findings in this report proved that hair follicles could be used to produce pluripotent stem cells and suggested that the genetic and micro-environmental elements of hair follicles might trigger higher and more efficient hair follicles re-differentiation.

Corresponding author
Seng Chiew Gan, gansc@utar.edu.my

# INTRODUCTION

Hair follicles are considered as a mini-organ which is important in animals and humans for protection from coldness (*Marioka, 2006*). Hair follicles are appendages which reside in the dermis in which its outgrowths through the epidermis form the whole hair (*Aasen*

*& Belmonte, 2010*). The continuous growth of hair occurs in three phases, namely anagen (growing), catagen (regression), and telogen (resting) (*Yoo et al., 2007*).

Genetic or environmental factors can result in alopecia or hair loss in many people (*Yoo et al., 2010*). Despite that hair disorders are not life threatening, hair still play a significant role in human from aesthetics and social perspectives (*Randall & Botchkareva, 2008*). The treatments for alopecia patients include drug therapy or hair transplantation. Drugs are only effective in mild cases of alopecia, and are often accompanied by unwanted side effects. Hair transplantation involves re-distributing existing hair follicles to bald areas. The procedure is very tedious, lengthy and bloody (*Yoo et al., 2010*). Hence, stem cell-based therapy may be employed, as minute number of stem cells could be harvested and expanded in the laboratory for hair transplantation.

Since the discovery of human induced pluripotent stem cells (hiPSC) in 2007, several studies have reported the successful generation of pluripotent cells from human hair follicles keratinocytes (*Aasen et al., 2008*; *Streckfuss-Bomeke et al., 2013*; *Novak et al., 2010*). All of these studies have successfully generated iPSC to study reprogramming efficiency and also to produce cardiomyocytes from the iPSC. Keratinocytes are the main cell type in the hair follicles (*Aasen et al., 2008*) and made up the surface of hair which acts as a protective barrier against external environment (*Aasen et al., 2008*). Primary keratinocytes exhibited limited propagation and required cultivation on irradiated 3T3 feeder layer (*Limat & Noser, 1986*). Despite the improvement in the culture-expansion protocol of keratinocytes (*Yoo et al., 2007*), the generation of pluripotent cells from keratinocytes offers an interesting and promising potential for making new hair follicles. This is because iPSCs has unlimited pro-liferation ability and pluripotent in nature. The keratinocyte which is newly re-programmed may maintain its tendency to differentiate into the same lineage upon exposure to the right induction microenvironment. Hence, keratinocyte-derived iPSC may be a suitable cell source for further re-generation of keratinocytes.

In this study, we aimed to generate iPSC from human hair follicular keratinocytes (HH-FKs) and determine the capability of the derived-iPSC to re-differentiate into keratinocytes. Generation of iPSC from HHFKs was carried out by performing transduction with OSKM factors. The generated iPSCs were characterized to evaluate the expression of pluripotency markers and its ability to form three germ layers. Upon exposure to keratinocytes differenti-ation induction medium, iPSCs-derived HHFKs expressed cytokeratin 14 (CK14), a marker of keratinocyte. From our data, we showed that iPSC derived from the human hair follicle keratinocytes was pluripotent based on the characterization performed and the iPSC could be re-differentiated back into keratinocytes. The differentiated keratinocytes represents new and valuable cell source to treat hair loss or skin regeneration in life-threatening burn cases.

## MATERIALS & METHODS

### Preparation of OSKM plasmid for viral transduction

Transformed *E. coli* encoding pMX-GFP, pMXs-hOCT3/4, pMXs-hSOX2, pMXs-hKLF4, pMXs-hc-MYC plasmids were stored in glycerol stock. The transformed *E. coli* consists of ampicillin resistant gene and was used as a marker for selection. A single colony of bacteria

was picked and dipped into the Luria-Bertani (LB) broth. The LB broth was incubated overnight at 37 °C with agitation. The bacteria suspension in the broth was subjected to plasmid extraction. The DNA plasmids were extracted according to the protocol from QIAGEN's Plasmid Midi Kit manual (Cat. No: 12143).

## Transfection of phoenix cells with plasmid DNA

Phoenix cells were used as packaging cells to produce retroviral carrying the OSKM gene. DNA plasmids (6.6 μg) were used to transfect the Phoenix cells. Phoenix cells were originated from Dr. Nolan's laboratory which was kindly donated by Dr. Kenneth Raj from Health Protection Agency (HPA), United Kingdom. The plasmids were transfected using calcium phosphate method. The plasmids were added to the 8.3 μl of 2.53 M calcium chloride. Then, the plasmids and calcium chloride were added to 2× HBS while mixing. The mixture was top up to 83 μl with distilled water.

The GFP signal was used as an indicator for the transfection efficiency. The time point at which the GFP signal was strongest would represent the optimum uptake of plasmids by the cells. The supernatants were collected at 72 h and 96 h when strongest GFP signals were observed. More than 80% of the total cells expressed GFP signals at these time points. The collected supernatants were filtered with 0.45 μm PVDF membrane before being used for transduction.

## Transduction of human hair follicle keratinocytes with retroviral supernatant

The HHFK (ScienCell, US) was seeded at 50,000 cells per well in a 6-well plate before transduction. The culture medium was removed from each well after 24 h. Freshly filtered retroviral supernatants were added into each well, followed by the addition of Polybrene (8 μg/ml). The plate was then centrifuged at 700 × g for 45 min at 32 °C. After that, the cells were incubated in a $CO_2$ incubator for 20 min. The supernatants were removed and washed twice with phosphate buffered saline (PBS) before being replaced with keratinocyte serum free medium (ScienCell, US). After 24 h, the cells were again transduced using the same protocol. A day after second infection, the cells were trypsinized and centrifuged at 400 × g for 5 min. The infected cells were resuspended in hESC culture medium supplemented with 1 mM valproic acid and plated onto wells treated with 0.1% gelatin and Mitomycin-C inactivated mouse embryonic fibroblasts as feeder layer. The cells were kept in a hypoxic incubator supplied with 5% oxygen. After two days, the culture medium was changed daily for 10 days. Following that, the cells were incubated in hESC culture medium without valproic acid in a normoxic incubator. The culture medium was changed daily until selection of potential iPSC-like colonies between day 14 and 28 for further cell expansion and characterisation. To study transduction efficiency, HHFK was transduced with a retrovirus carrying GFP gene to determine the transduction efficiency. The transduction efficiency was calculated based on number of cells expressing GFP to the total cell number. The process of reprogramming until the iPSC clones were picked was showed in Fig. 1.

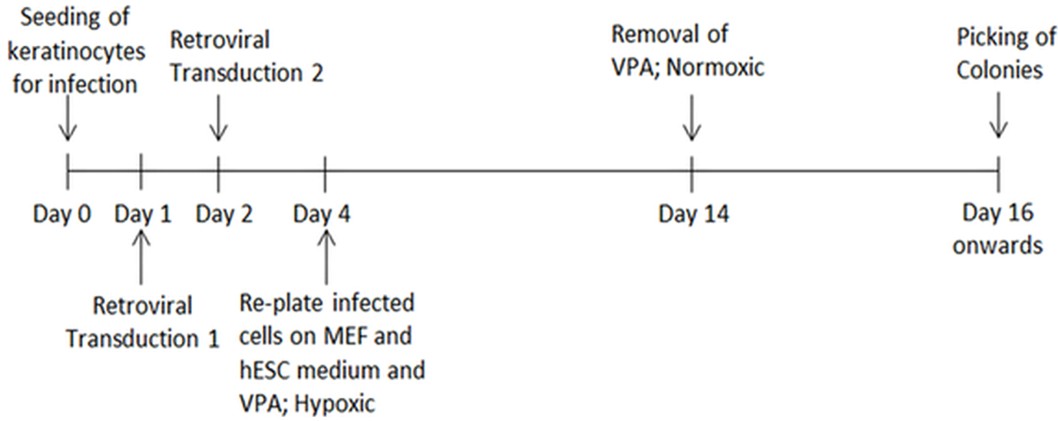

**Figure 1 The timeline showed the HHFK reprogramming process.** The timeline includes the process from culturing of the parental cells before initiation of the reprogramming process to picking of each iPSC clones for further expansion.

## Live staining of iPSC using SSEA4 markers

For SSEA4 live staining (Stemgent, Inc., Lexington, MA, USA), the primary antibody was diluted to a final concentration of 2.5 µg/ml in DMEM/F12 (Gibco, Big Cabin, OK, USA). The culture medium was first removed from the cell culture. The diluted primary antibody was then added and the culture was incubated for 30 min at 37 °C and 5% $CO_2$. Following that, the primary antibody was removed and the culture was washed with DMEM/F12 gently for two times. Fresh culture medium was added and the cells were examined under the fluorescent microscope using a phycoerythrin filter. The protocol for SSEA4 live staining was performed according to the manufacturer's manual.

## Characterisation of iPSC using immunofluorescence staining

The cells were fixed with 4% paraformaldehyde for 10 min at room temperature. After this and each of the following steps, the cells were washed with 1X PBS for three times. Next, the cells were treated with 0.1% bovine serum albumin in PBS for 15 min at room temperature. For intracellular staining, the cells were treated with BD Perm/Wash buffer (Becton Dickinson, US) before proceeding with antibody staining. Next, the cells were stained with fluorescent-conjugated antibody for 1 h at room temperature or overnight at 4 °C. The cells were counter-stained with DAPI for 30 min at room temperature to stain the nucleus. The antibodies used were: SSEA4 (1:50), TRA-1-60 (1:50), TRA-1-81 (1:50), Nanog (1:50), Sox2 (1:50), and Oct4 (1:50). All antibodies were purchased from Becton Dickinson.

## Characterisation of iPSC using flow cytometric analysis

For surface marker staining, $2 \times 10^5$ cells per sample were aliquoted into a 5-ml FACS tube. The cells were centrifuged at $300 \times g$ for 5 min. While centrifuging, 100 µl of antibody mixes were prepared at optimal working dilutions. After centrifugation, the supernatants were removed without disturbing the cell pellets and the cells were resuspended in the appropriate antibody mixes. The tubes were gently tapped to mix the antibodies and cells.

The tubes were incubated for 60 min at room temperature in dark. One ml of 1% BSA in PBS was then added to each of the tubes. The tubes were spun at $300 \times g$ for 5 min. The supernatants were removed and the cells were resuspended in 200 µl of FACS Flow (Becton Dickinson, Franklin Lakes, NJ, USA). The samples were analysed by flow cytometry as soon as possible. The antibodies used were: SSEA4 (1:50), TRA-1-60 (1:50), and TRA-1-81 (1:50). All antibodies were purchased from Becton Dickinson.

For intracellular staining, $2 \times 10^5$ cells per sample were aliquoted into a 5-ml FACS tube. The cells were centrifuged at $300 \times g$ for 5 min. Following that, the cells were fixed in 250 µl of 4% paraformaldehyde for 10 min. One ml of 1% BSA in PBS was then added to each of the tubes. The tubes were gently mixed and spun at $300 \times g$ for 5 min to wash the cells. The supernatants were completely removed without disturbing the cell pellets and the cells were then resuspended in 500 µl of 1X BD Perm/Wash buffer. The tubes were mixed by gentle tapping and incubated at room temperature for 15 min. Then, the tubes were spun at $300 \times g$ for 5 min. The 1X BD Perm/Wash buffer in 1% BSA/PBS was used as a diluent for the antibodies and wash buffer for the remaining steps. The protocol was continued according to the methods stated in the above paragraph. The antibodies used were: NANOG (1:50), SOX2 (1:50), and OCT4 (1:50). All antibodies were purchased from Becton Dickinson.

## Characterisation of iPSC using reverse-transcription polymerase chain reaction (RT-PCR)

Total RNA was extracted from iPSC using the GeneAll Ribospin RNA extraction kit (GeneAll, Seoul, South Korea). The cDNA was synthesized from 500 ng to 1 µg of RNA using 5X iScript Reverse Transcription Supermix kit (Bio-Rad, Hercules, California, USA). The conversion protocol and reaction setup were according to the given protocol by the manufacturer. The PCR was performed using the reagents from MyTaq™ Red Mix (Bioline, London, UK). The primers used were *SOX2*, *OCT4*, *KLF4*, *C-MYC*, *NANOG*, *LEFTY*, and *GABRB*. Each PCR products were electrophoresed in 1.5% agarose gel to check for its DNA band size. The bands were stained with GelRed (Biotium Inc., Hayward, CA, USA) and visualised under an ultraviolet transilluminator.

### *In vitro* differentiation for three germ layers formation

The iPSCs were plated for mesoderm differentiation on dishes coated with matrigel (Becton Dickinson, New Jersey, USA). The iPSCs were grown to 70% confluency before differentiation. For mesoderm differentiation, the culture medium was changed to mesoderm differentiation medium (R & D Systems; Minneapolis, MN, USA). The differentiation medium was changed again after 16 h. The differentiated cells were harvested for immunofluorescence staining after 36–48 h after initial treatment. For ectoderm and endoderm differentiation, the iPSCs were first harvested and transferred to a non-cell culture treated well. The medium used was hESC culture medium without fibroblast growth factor 2 (FGF2). The iPSCs formed embryoid bodies (EBs) the next day and the EBs were cultured in the same medium for five days. The medium was changed every two days. After five days, the EBs were used for ectoderm and endoderm differentiation. For ectoderm and endoderm differentiation, the cell treatments with specific growth factors were continued

for five days with change of medium every 48 h. For ectoderm differentiation, the growth factor used was noggin with a concentration of 100 ng/ml. For endoderm differentiation, activin A was added at a concentration of (50 ng/ml).

## Directed differentiation into keratinocytes

The iPSCs were plated onto dishes pre-coated with matrigel. After 24 h, cells were treated with keratinocytes serum free medium (ScienCell, US) containing retinoic acid (RA) (concentration of 25 ng/ml) and bone morphogenetic protein-4 (BMP-4) (concentration of 1 μM). The culture medium with growth factors was changed every 48 h for a minimum of 15 days before the cells were harvested for immunofluorescence staining and RT-PCR.

## Characterisation of keratinocytes derived from iPSCs using immunofluorescence staining

The keratinocytes were fixed with 4% paraformaldehyde for 10 min at room temperature. After each of the following steps, the cells were washed with PBS for three times to remove the reagents leftover. Next, the cells were treated with 0.1% bovine serum albumin in PBS for 15 min at room temperature. The cells were treated with BD Perm/Wash buffer before proceed with antibody staining. Next, the cells were stained with fluorescent-conjugated antibody for 1 h at room temperature or overnight at 4 °C. The cell nuclei were stained with DAPI for 30 min at room temperature. The antibody used for differentiated cells was CK14 (1:50) (abD Serotec; North Carolina, USA).

## Characterisation of keratinocytes derived from iPSCs using RT-PCR

The cDNA was synthesized using the total RNA extracted from iPSC using 5X iScript Reverse Transcription Supermix kit (Bio-Rad, Hercules, California, USA). The PCR was performed using the reagents from MyTaq$^{TM}$ Red Mix (Bioline, London, UK). The primer set used was forward: 5′GACCATTGAGGACCTGAGGA and reverse: 5′CATACTTGGTGCGGAAGTCA to detect CK14 (band size of 113 base pair) gene expression. The PCR product was then run in 1.5% agarose gel and viewed under an UV transilluminator.

# RESULTS

## Reprograming of HHFK and its efficiency

The morphology of iPSC resembles the morphology of hESC in terms of a compact colony with large nucleoli, large nucleus to cytoplasm ratio, distinct cell border, and slightly rounder in shape (Fig. 2A). The transduction efficiency of HHFK using the retrovirus carrying OSKM transcription factors was 60%. Figure 2A showed that the iPSC clones showed pseudo red PE signal for SSEA4. This showed that the generated iPSC clones were expressing SSEA4.

In this study, 34 clones were selected for further expansion. However, only three clones were selected for further characterization and differentiation as some clones were not able to survive during the expansion process. The clones were initially cultured on feeder layer up to 26 passages. At the same time, the feeder based clones were switched to feeder-free culture system starting from as early as passage 9. The selected iPSC clones were able to

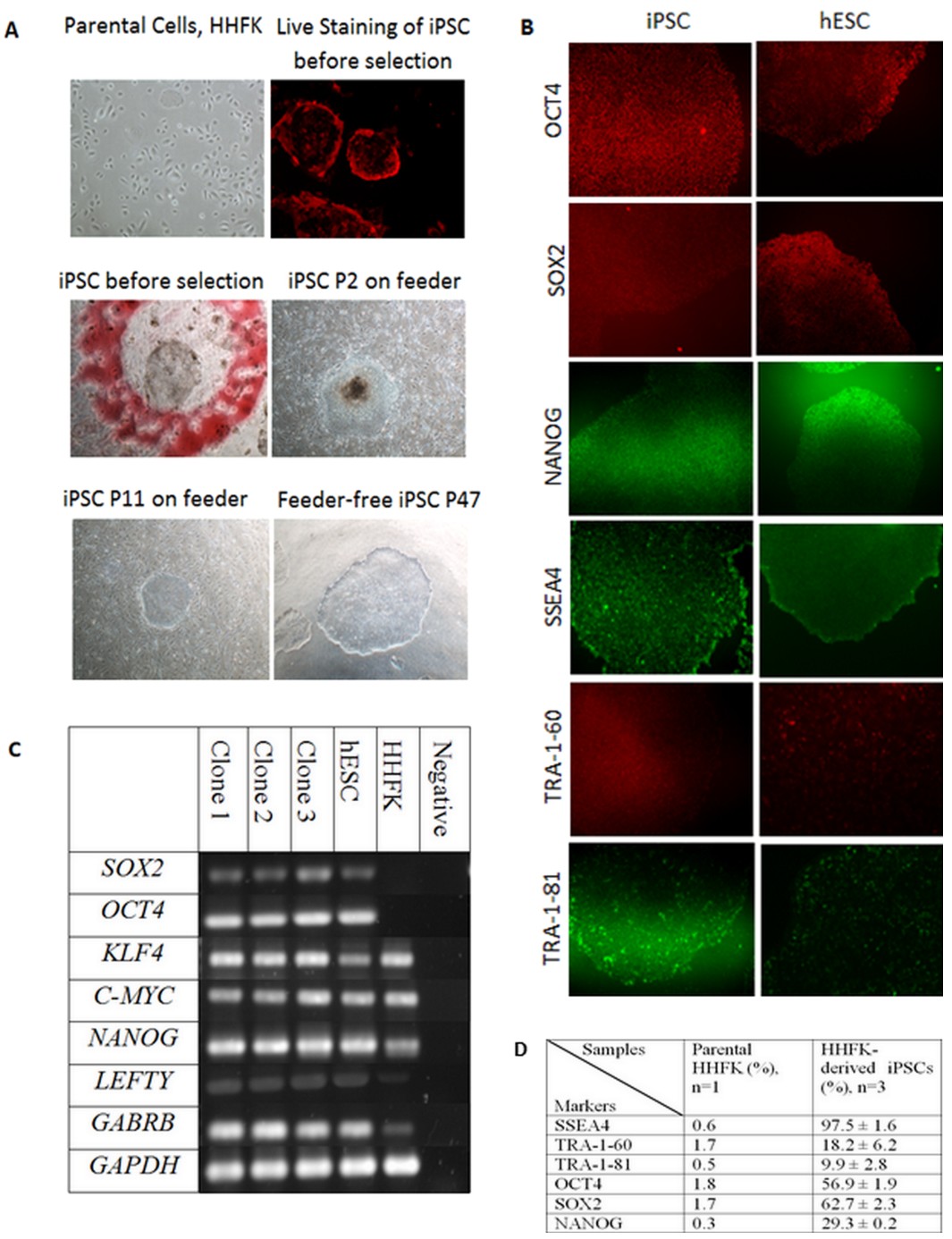

**Figure 2** **Reprogramming of Human Hair Follicular Keratinocytes (HHFKs) into iPSC and characterization of iPSC.** (A) (Parental cells, HHFK) The image of parental cells, HHFKs was before infection with viruses. (Live Staining of iPSC before selection) The iPSC colonies formed showed positive live-staining (red) for SSEA4, before selection. (iPSC before selection) The image showed a single iPSC clone (in red circle) before selection for further culture. (iPSC P2 on feeder & iPSC P11 on feeder) Image of iPSC, at passage 2 and 11, cultured on feeder layer. (Feeder-free iPSC P47) Image of iPSC colony which was grew in feeder-free system at passage 47. Magnification: 100×. (continued on next page...)

**Figure 2 (...continued)**
(B) Immunofluorescence staining of pluripotent markers for iPSCs and hESC. The pluripotency markers include OCT4, SOX2, NANOG, SSEA4, TRA-1-60, and TRA-1-81. Magnification: 100×. (C) The expression of pluripotency markers by iPSC clones, hESC and HHFK using RT-PCR. The pluripotent genes which were being analysed include SOX2, KLF4, C-MYC, NANOG, LEFTY, and GABRB. The housekeeping gene being used was GAPDH. (D) Comparison of expression of pluripotency markers in HHFKderived iPSCs and parental HHFK.

expand up to 47 passages in feeder free condition (Fig. 2A). All the iPSC clones grew in colonies and showed typical hESC morphology such as distinct border, and tightly packed cells. The iPSCs can be cultured without any replicative crisis and the culture was terminated after completion of the differentiation step.

The reprogramming efficiency was determined based on the number of colonies formed to the total initial number of cells seeded. The colonies were counted based on the morphology. We obtained a total of 218–372 iPSC colonies from each of the three separate experiments with the initial total seeded HHFK of 50,000 cells. The average reprogramming efficiency was 0.59% ($\pm$0.03%).

## Characterisation of iPSCs

After the selection and expansion of the iPSCs, the immunofluorescence staining and flow cytometry analysis were performed to check for the presence of pluripotent-associated proteins. The generated iPSC clones were cultured up to at least passage 10 before characterisation for its pluripotency markers. The iPSC clones were stained with antibodies against OCT4, SOX2, NANOG, SSEA4, TRA-1-60, and TRA-1-81.

The immunofluorescence staining showed that the iPSC clones expressed intracellular pluripotency marker such as OCT4, SOX2, and NANOG (Fig. 2B) as well as the pluripotency surface marker such as SSEA4, TRA-1-60, and TRA-1-81 (Fig. 2B). The iPSC clones were further characterised using flow cytometry to check for the percentage of cells that express the pluripotent markers. As a comparison, both hESC (BGO1V cell line) and parental cells (HHFK) were included.

SSEA4 expression was significantly higher in HHFK-derived iPSC compared to HHFK, with the percentage of 97.5 $\pm$ 1.6 and 0.6, respectively. TRA-1-60 (18.2 $\pm$ 6.2) and TRA-1-81 (9.9 $\pm$ 2.8) were also significantly expressed by HHFK-derived iPSC compared to parental cells (1.7 and 0.5, respectively). OCT4 expression significantly increased from 1.8 in parental cells to 56.9 $\pm$ 1.9 in iPSCs. SOX2 expression also showed significant increased from 1.7 in parental to 62.7 $\pm$ 2.3 in HHFK-derived iPSCs. Lastly, for NANOG expression, the percentage in parental cells (0.3) also significantly increased compared to the percentage in iPSCs (29.3 $\pm$ 0.2).

RT-PCR was performed to identify the gene expression of pluripotent markers in the iPSCs, hESC, and parental cells. GAPDH was used as the housekeeping gene for control. *SOX2*, *OCT4*, *KLF4*, *C-MYC*, *NANOG*, *LEFTY*, and *GABRB* were used as the pluripotent markers to check for its presence in iPSCs, hESC, and HHFK. All three clones (Fig. 2C) showed comparable expression of pluripotent markers between the clones. All three clones also showed comparable expression of all pluripotent markers to the hESC. For the parental

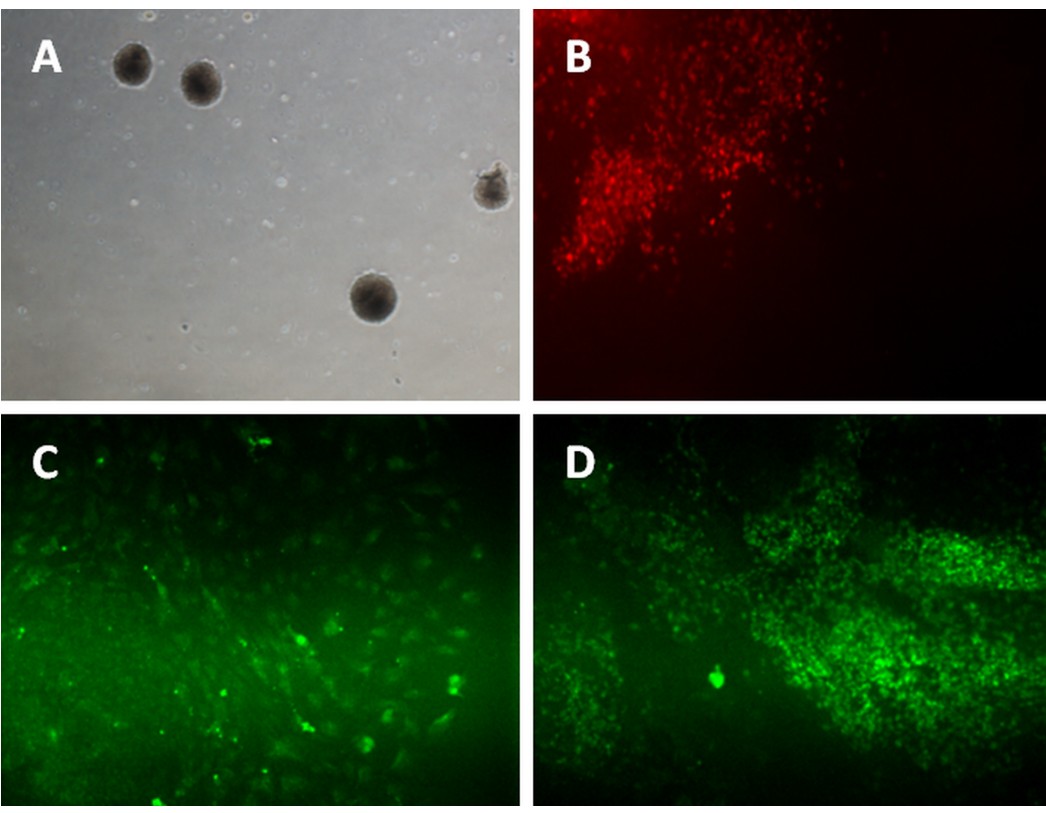

**Figure 3 Differentiation of iPSC into three germ layers.** Formation of embryoid bodies before the differentiation process (A). The mesoderm lineage was stained against BRACHYURY (B), ectoderm was stained against MAP2 (C), and endoderm lineage was stained against SOX17 (D). Magnification: 100×.

cells, there was presence of certain pluripotent markers such as *KLF4*, *C-MYC*, *NANOG*, *LEFTY*, and *GABRB*. However, expression of *SOX2* and *OCT4* were not detected (Fig. 2C).

The iPSCs were differentiated into mesoderm lineage using the monolayer method. The cells were cultured to reach 70% confluency on matrigel-coated plates before induction. Two days after differentiation induction, the iPSCs and hESC were stained with Brachyury, a marker for mesoderm lineage. Figure 3B showed the iPSC and hESC were able to form mesoderm lineage cells that expressed Brachyury.

For ectoderm and endoderm lineage differentiation, the iPSCs and hESC were cultured on uncoated multi-well plate for five days before being transferred to 0.1% gelatin-coated plate. The subcultured iPSC and hESC colonies curled and formed a round-shaped structure called embryoid bodies (Fig. 3A). Cells started to grow out from attached EBs. Activin A was added for endoderm lineage differentiation while Noggin was added for ectoderm lineage differentiation. Figure 3C showed the cells expressing MAP2 marker, which is an ectoderm marker. Figure 3D showed expression of Sox17 in iPSC and hESC. SOX17 is a marker for endoderm lineage.

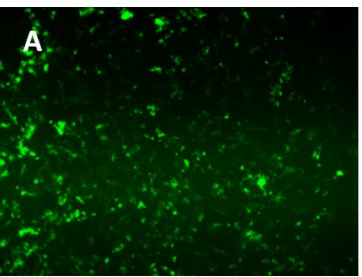
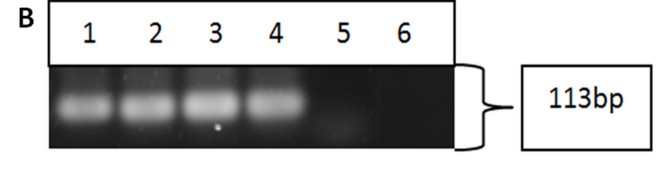

**Figure 4 Characterisation of differentiated cells using IF staining and RT-PCR.** The differentiated cells were stained against cytokeratin 14 marker. The differentiated cells showed expression for CK14 marker (A). Magnification: 100×. The differentiated cells from iPSCs and hESC showed the expression of cytokeratin 14. (Lane 1) Clone 1, (Lane 2) Clone 2, (Lane 3) Clone 3, (Lane 4) hESC, (Lane 5) HHFK, and (Lane 6) Negative Control (B).

## Differentiation of iPSCs into keratinocytes

Upon successful verification of iPSC clones, the cells were re-differentiated back into keratinocytes in the presence of bone morphogenetic-4 protein and retinoic acid. After 15 days, the cells were harvested and subjected to immunofluorescence staining and reverse-transcription polymerase chain reaction to check for the cytokeratin 14 marker, which is a typical marker for basal keratinocytes. Both the differentiated cells from iPSC (Fig. 4A) and hESC showed positive staining for CK14. Figure 4B showed that the three iPSC clones and hESC also expressed CK14 gene.

## DISCUSSION

Keratinocytes from human hair follicles were used as the candidate cells for reprogramming as it is easy to be obtained by simple plucking. Thus, this cell type offers advantage over other cell types in a patient deemed unsuitable to carry out a skin biopsy or bone marrow aspirate (*Streckfuss-Bomeke et al., 2013*). In this study, iPSCs have been successfully generated from HHFK using a retrovirus encoding four types of transcription factors (*OCT4*, *SOX2*, *KLF4*, and *C-MYC*). The factors were delivered at the same time using a single factor vectors. This method was among the first and most established method for generation of iPSCs as different type of cells (such as fibroblasts, neural stem cells, adipose cells, and amniotic cells) have been used to generate iPSCs successfully (*Takahashi et al., 2007*; *Robinton & Daley, 2012*). This method also reported higher reprogramming efficiency compared to other viral transduction methods using adenovirus or Sendai virus (*Oh et al., 2012*).

There are several factors such as cell source, oxygen level of cell incubation and different infection systems which have been known to affect the reprogramming efficiency. Here, we reported a 59-fold increase in reprogramming efficiency in HHFK (0.59%) compared to human fibroblasts reprogramming (0.01%) (*Takahashi et al., 2007*). Keratinocytes used in this study is a type of highly proliferative epithelial cells with limited doubling passage (*Aasen & Belmonte, 2010*). In iPSCs generation, fibroblast cells need to undergo mesenchymal-to-epithelial transition (MET), thus requiring a bigger effort for the transition and this resulted in lower efficiency (*Aasen et al., 2008*). Meanwhile, keratinocytes could skip the MET process and thus resulted in a higher reprogramming efficiency in this study compared to

those using fibroblasts (*Takahashi et al., 2007*). Reprogramming under hypoxic condition in human cells was observed to enhance the reprogramming efficiency by 4 fold (*Gonzalez, Boue & Belmonte, 2011*). It is noteworthy that the reprogramming efficiency of retrovirus transduction in this study exceeded the reprogramming efficiency of keratinocytes (0.03%) using polycistronic lentivirus (*Novak et al., 2010*).

The inclusion of *OCT4* and *SOX2* in this study as the transcription factors for reprogramming is rather important (*Takahashi et al., 2007*; *Yu et al., 2007*). The *OCT4/SOX2* motif was found located at the upstream of the Nanog transcription start site of *NANOG* gene (*Pan & Thomson, 2007*). *OCT4* would form heterodimer with *SOX2* and further enhanced *NANOG* expression which was already present in HHFK used in our study (Fig. 4). The expression of *OCT4*, *SOX2* and *NANOG* in iPSCs will promote the maintenance of pluripotency of iPSC and suppress the genes involved in differentiation (*Boyer et al., 2005*; *Pan & Thomson, 2007*). In contrary to our study, HHFK cells used to generate iPSCs in *Novak et al. (2010)* did not show presence of *NANOG*. Nevertheless, *NANOG* expression has been reported in other human keratinocytes cell line such as HaCat, and keratinocytes stem cells derived from the bulge of human hair follicles (*Yu et al., 2006*; *Palla et al., 2015*). Therefore, there is a possibility that HHFK consists some keratinocytes stem cells which expressing NANOG. However, the mRNA level might be too low in our study to produce a detectable protein level (Fig. 2C). Various processes are involved in the regulation of protein abundance such as post-transcriptional modification and degradation of mRNAs and proteins which could affect the protein level (*Vogel & Marcotte, 2012*).

We also believed that the inclusion of *C-MYC* as one of the reprogramming factors will enhance the reprogramming efficiency (*Nagakawa et al., 2008*). *C-MYC* could act on miR-21 and miR-29a by down-regulating the level of these two microRNAs (*Yang, Li & Rana, 2011*). The inhibition of these two mircoRNAs would inhibit the p53 and ERK1/2 pathways (*Yang, Li & Rana, 2011*), leading to an increase in the reprogramming efficiency (*Yang, Li & Rana, 2011*). As *C-MYC* has a role as an oncogene, many reprogramming studies have eliminated *C-MYC* from their reprogramming cocktails (*Okita & Yamanaka, 2012*; *Piao et al., 2014*).

*KLF4* is involved in suppression of early somatic genes such as *TGFB1*, *PDGFRA*, and *COL6A1* and activation of pluripotency genes (*Polo et al., 2012*). *KLF4* binds to *OCT4* and *SOX2* via the C-terminus which contain three zinc fingers to form a complex. *KLF4* and *OCT4* binding site are also present in the *NANOG* promoter. Introduction of mutant *KLF4* would reduce reprogramming efficiency (*Wei et al., 2009*). In this study, the HHFK was identified expressing endogenous *KLF4* and *C-MYC* (Fig. 4). *KLF4* is still needed for reprogramming although endogenous *KLF4* was reported to be highly expressed in keratinocyte (*Aasen et al., 2008*). The dermal papilla (DP) cells isolated from hair follicles also expressed endogenous *SOX2*, *KLF4*, and *C-MYC* (*Tsai et al., 2011*). Reprogramming of the dermal papilla cells with the use of only one factor, *OCT4* has successfully generated iPSCs but with lower efficiency of 0.088% and the colonies only appeared three weeks later (*Tsai et al., 2011*).

Combination of surface markers and morphology identification is required to confirm the selection of true iPSC. We observed that the generated iPSCs were of true identity via

expression of hESC-like surface and intracellular markers such as TRA-1-60, TRA-1-81, SSEA4, OCT4, SOX2 and NANOG. Characterisation of iPSC using the surface markers SSEA4, TRA-1-81 and TRA-1-60 was performed at the protein level using the flow cytometric and immunofluorescence analysis. The results in this study showed comparable expression of the above mentioned surface markers for iPSC with that of hESC. For TRA-1-60 and TRA-1-81, the percentage of expression was lower in iPSC compared to hESC (Fig. 2D). However, the SSEA4 level was higher in iPSC than hESC (Fig. 2D). Previous studies have not report on the differences in the level of SSEA4 between iPSC and hESC and association between the level differences. SSEA4 is the carbohydrate epitopes of the globo-series glycolipids (Zhao et al., 2012). SSEA4 are found in mesenchymal stem cells, neural progenitor cells and corneal epithelial cells (Truong et al., 2011) apart from their presence in oocytes, zygotes and early cleavage-stage embryos (Zhao et al., 2012). SSEA4 and SSEA3 usually are expressed before TRA-1-60 and TRA-1-81 (Chan et al., 2009).

The iPSCs were able to be differentiated *in vitro* to form the three embryonic germ layers, namely, ectoderm, endoderm, and mesoderm (Fig. 3). Following that, the clones of iPSC were able to be differentiated into keratinocytes that expressed the cytokeratin 14 (CK14) (Figs. 4A and 4B). CK14 expression in hair follicle keratinocytes has been reported in other studies (Gho et al., 2004; Novak et al., 2010). Two growth factors, retinoic acid (RA) and bone morphogenetic protein 4 (BMP4) were used to direct the differentiation of iPSCs into keratinocytes. RA directs the iPSCs into epithelial fate while BMP-4 suppresses the differentiation of the reprogrammed cells into neural lineage (Metallo et al., 2008). In addition, the RA-induced differentiation offered an efficient way to generate keratinocytes with higher proliferative capacity (Metallo et al., 2008).

To our knowledge, there has not been any published study on keratinocytes reprogramming using three or lesser factors. Thus, study to optimize the reprogramming method to facilitate the successful generation of keratinocytes using three factors or lesser is needed. By reducing the number of reprogramming factors, the risk of cancer formation and genomic integrations can be reduced (Tsai et al., 2011). This could be coupled with the use of small molecules to enhance the reprogramming efficiency (Oh et al., 2012). With the high keratinocytes reprogramming efficiency, keratinocytes can represent one of the non-invasive and easily accessible cell sources for reprogramming.

In this study, we successfully demonstrated the potential of hair follicle keratinocytes to be used as a cell source for reverse differentiation and further re-differentiation into keratinocytes. Although balding is not considered as a life-threatening disease and the majority cases are due to the aging process, it still imposes an immense psychological influence on the affected individual (Wells, Willmoth & Russell, 1995). Balding can usually be reduced by hair transplantation or drug treatment, which is also accompanied by many drawbacks (Martin & Mangubat, 2000; Harries et al., 2010). The current research demonstrated that the pluripotent cells can be expanded *in vitro* and used for more generation of keratinocytes. The keratinocytes may be used to form hair follicles in hair transplantation. Other than that, the keratinocytes may be indicated to repair wound on burn patients by direct seeding onto affected site (Mcheik et al., 2014).

Since the iPSCs were successfully generated from human hair follicular keratinocytes, future work should focus on generating iPSCs-derived HHFK using relatively safer method to eliminate transgene integration into host genome. This safer cell source represents valuable cells for clinical use and most importantly, generation of patient-specific cells. These autologous cells can be used for the patient with less issue of rejection. In contrast, using hESC-derived cell source for transplantation will result in tissue rejection (*De Almeida et al., 2013*). With the improvement in the differentiation protocol, keratinocytes can be derived from the patient's iPSCs. This keratinocytes source represents a valuable source for *in vitro* hair cloning and used to treat hair balding or for grafting of burn patients.

## CONCLUSION

In conclusion, iPSCs have been successfully generated from HHFKs using retroviral transduction method. The four transcription factors used were *OCT4*, *SOX2*, *KLF4*, and *C-MYC*. The iPSCs generated expressed hESC markers such as TRA-1-60, TRA-1-81, SSEA4, *OCT4*, *SOX2*, *NANOG*, *LEFTY*, and *GABRB*. The iPSCs were able to be differentiated *in vitro* to form the embryonic three germ layers, namely, ectoderm, endoderm, and mesoderm. From the HHFK-derived iPSCs, it was able to be differentiated into keratinocytes expressing CK14. The differentiated keratinocytes from iPSC could represent a new valuable source for *in vitro* hair cloning or use to treat hair balding or for grafting in burn patients.

## ACKNOWLEDGEMENTS

The authors would like to thank Dr. Shigeki Sugii [A⋆ STAR, Singapore Bioimaging Consortium (SBIC)] for the transformed *E. coli* and Dr. Kenneth Raj (Health Protection Agency, UK) for providing the Phoenix cells.

### Funding

University Tunku Abdul Rahman Research Fund (UTARRF) nos. 6200 G07, G10 and G11 funded the research. The funders had no role in study design, data collection and analysis, decision to publish, or preparation of the manuscript.

### Grant Disclosures

The following grant information was disclosed by the authors:
University Tunku Abdul Rahman Research Fund (UTARRF): Nos. 6200 G07, G10 and G11.

### Competing Interests

The authors declare there are no competing interests.

### Author Contributions

- Sheng Jye Lim performed the experiments, analyzed the data, wrote the paper, prepared figures and/or tables.

- Shu Cheow Ho performed the experiments, analyzed the data.
- Pooi Ling Mok, Kian Lee Tan and Alan H.K. Ong conceived and designed the experiments, reviewed drafts of the paper.
- Seng Chiew Gan conceived and designed the experiments, analyzed the data, contributed reagents/materials/analysis tools, wrote the paper, reviewed drafts of the paper.

## Data Availability

The raw data has been supplied as Data S1.

## Supplemental Information

Supplemental information for this article can be found online at http://dx.doi.org/10.7717/peerj.2695#supplemental-information.

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
