# Peer review of "Induced pluripotent stem cells from human hair follicle keratinocytes as a potential source for in vitro hair follicle cloning"

_PeerJ, doi:10.7717/peerj.2695_

## Round 0.1 · original submission · Major Revisions

Please follow the reviewer's comments to correct your manuscript carefully.

Reviewer 1 ·

Basic reporting

The submission follows to PeerJ policies. However, it does not include sufficient introduction about the other manuscript that employs human hair follicle keratinocytes to be reprogrammed to iPS. There are 13 articles where this technology was employed. They should reference this relevant prior literature and establish the main aim by the end of the introduction to understand the proposal of the work done.

Experimental design

The submission describes an original research since they propose iPS obtained from keratinocytes as a source back of keratinocytes. Most of the prior papers found keratinocytes reprogramming useful to reach cardiomyocytes or neuronal cells. The research is well conducted and the methodology employed suitable and novel. Still, they have to strongly indicate the gap being investigated and to how the study contributes to filling that gap.

Validity of the findings

The findings found in the manuscript are correct and they are supported by well-conducted experiments. However, the figures are not clear enough. Higher magnification images to see whether the signal is specifically located inside cells should be included. In addition, Figure 6 should be consistent with Figure 5 (labelling of RT-PCR gels). It was found an overexpression of NANOG and KLF4 in HHFK cells since these are cells were purchased from a company and they claim that they are keratinocytes fully differentiated; this result should be carefully explained. The conclusions should be appropriately connected to the original question investigated.

Additional comments

In general, the article is correct and employs a novel technology that could be useful in the future to treat hair loss, although this is still a speculation and animal experiment should be carried out in the future. The weakness of the article is the absence of a clear main aim at the beginning and the quality of the images that should be improved since all their conclusions are base on immunohistochemistry assays.

Reviewer 2 ·

Basic reporting

Actually, quality of figures needs to be improved and more organized data representation is needed.

Experimental design

For iPSC generation, they used retrovirus. Authors are recommended to use the more updated iPSC generation technique for generation of exogenous DNA integration-fee iPSC, which will be more safe and efficient for further study.
More markers are needed for the efficient differentiation into hair keratinocytes.

Validity of the findings

Quality of figures needs to be improved and more organized data representation is needed.

Additional comments

Actually, quality of figures needs to be improved and more organized data representation is needed. For iPSC generation, they used retrovirus. Authors are recommended to use the more updated iPSC generation technique for generation of exogenous DNA integration-fee iPSC, which will be more safe and efficient for further study.
More markers are needed for the efficient differentiation into hair keratinocytes.

---

## Round 0.2 · accepted · Accept

The quality of the revised version was significantly improved and it is acceptable for publication.